# Immune-Checkpoint Inhibitors in Advanced Bladder Cancer: Seize the Day

**DOI:** 10.3390/biomedicines10020411

**Published:** 2022-02-09

**Authors:** Brigida Anna Maiorano, Ugo De Giorgi, Davide Ciardiello, Giovanni Schinzari, Antonio Cisternino, Giampaolo Tortora, Evaristo Maiello

**Affiliations:** 1Oncology Unit, IRCCS Foundation Casa Sollievo della Sofferenza, 71013 San Giovanni Rotondo, Italy; d.ciardiello@operapadrepio.it (D.C.); e.maiello@operapadrepio.it (E.M.); 2Department of Translational Medicine and Surgery, Catholic University of the Sacred Heart, 00168 Rome, Italy; giovanni.schinzari@policlinicogemelli.it (G.S.); giampaolo.tortora@policlinicogemelli.it (G.T.); 3Department of Oncology, IRCCS Istituto Scientifico Romagnolo per lo Studio e la Cura dei Tumori (IRST) “Dino Amadori”, 47014 Meldola, Italy; ugo.degiorgi@irst.emr.it; 4Medical Oncology Unit, Department of Precision Medicine, “Luigi Vanvitelli” University of Campania, 80131 Naples, Italy; 5Comprehensive Cancer Center, Medical Oncology Unit, IRCCS Foundation “A. Gemelli” Policlinic, 00168 Rome, Italy; 6Urology Unit, IRCCS Foundation Casa Sollievo della Sofferenza, 71013 San Giovanni Rotondo, Italy; antonio.cisternino@operapadrepio.it

**Keywords:** urothelial carcinoma, bladder cancer, mUC, BCa, ICI, immune checkpoint inhibitor, platinum, avelumab, pembrolizumab, PD1

## Abstract

Background: In advanced bladder cancer (BCa), platinum-based chemotherapy represents the first-choice treatment. In the last ten years, immune checkpoint inhibitors (ICIs) have changed the therapeutic landscape of many solid tumors. Our review aims to summarize the main findings regarding the clinical use of ICIs in advanced BCa. Methods: We searched PubMed, Embase, and Cochrane databases, and conference abstracts from international congresses (ASCO, ESMO, ASCO GU) for clinical trials, focusing on ICIs as monotherapy and combinations in metastatic BCa. Results: 18 studies were identified. ICIs targeting PD1 (nivolumab, pembrolizumab), PD-L1 (avelumab, atezolizumab, durvalumab), and CTLA4 (ipilimumab, tremelimumab) were used. Survival outcomes have been improved by second-line ICIs, whereas first-line results are dismal. Avelumab maintenance in patients obtaining disease control with chemotherapy has achieved the highest survival rates. Conclusions: ICIs improve survival after platinum-based chemotherapy. Avelumab maintenance represents a new practice-changing treatment. The combinations of ICIs and other compounds, such as FGFR-inhibitors, antibody-drug conjugates, and anti-angiogenic drugs, represent promising therapeutic approaches. Biomarkers with predictive roles and sequencing strategies are warranted for best patient selection.

## 1. Introduction

Bladder cancer (BCa) accounts for about 3% of tumors, being counted among the ten most common cancer subtypes and the thirteenth leading cause of cancer-related death worldwide [1,2]. BCa incidence is around 15 cases/100,000 inhabitants/year in the US and Europe [2,3]. Main risk factors for this malignancy include tobacco smoking, exposure to chemicals, family history, and male sex [4]. BCa is around four times more common among the male than the female population. This is likely attributable to more tobacco use and far more working exposure to chemical agents among men, and similarly explains the rising BCa incidence among women in developed countries [5]. The frequency of BCa rises with age, and over 90% of diagnoses occur in >55 year-old people [1,2]. Over 50% of cases are diagnosed in situ. The 5-year survival rate is 95.8%. For metastatic disease, the 5-year survival rate is <5% [1]. Most frequently, BCa has features of urothelial carcinoma (UC) [6]. Over the last few decades, chemotherapy has represented the first-choice treatment in advanced disease. In the first line, cisplatin-based combination regimens are recommended for eligible patients. Carboplatin is an alternative option for cisplatin-ineligible subjects, representing around 1/3 of BCa patients (creatinine clearance <60 mL/min, or Eastern Cooperative Oncology Group-Performance Status (ECOG-PS) <2, or hearing loss >grade 2 (G2), or peripheral neuropathy >G2, or symptomatic heart failure per New York Heart Association (NYHA) stage III/IV) [7,8,9]. First-line platinum-based chemotherapy results in objective response rates (ORR) of around 65–75%, progression-free survival (PFS) of 6–8 months, and overall survival (OS) of around 12–15 months with cisplatin and 6–9 months with carboplatin [10,11,12,13]. However, platinum failure is often associated with significant physical function impairment and quality of life deterioration, causing only a minority of patients to reach second and later lines of therapy (around 35–40%) [14,15]. After platinum failure, single-agents chemotherapy is mostly used, with a median survival of around 6 months, and ORR <10% [14,15,16,17].

Immunotherapy with immune checkpoint inhibitors (ICIs) has significantly changed the therapeutic strategies for many solid tumors in the last ten years, BCa included [18,19]. ICIs are monoclonal antibodies targeting cell surface proteins programmed death-1 (PD1), programmed death-ligand 1 (PD-L1), or cytotoxic T-lymphocyte—associated antigen 4 (CTLA-4), interrupting their interactions with specific ligands and removing the inhibition of T cells, thereby activating their cytotoxicity [18]. In May 2016, the Food and Drug Administration (FDA) approved the first ICI for BCa: atezolizumab, an anti-PD-L1 agent. Subsequently, other ICIs targeting PD-L1 (avelumab and durvalumab) or PD1 (nivolumab and pembrolizumab) improved survival over chemotherapy in pretreated patients [7,8]. In the US, pembrolizumab and atezolizumab have been authorized as first-line treatments in platinum-ineligible PD-L1^+^ BCa patients [7,8,20]. In 2020, the FDA granted the breakthrough approval to the first-line combination of pembrolizumab and the antibody-drug conjugate (ADC) enfortumab vedotin in cisplatin-ineligible patients, and avelumab was approved by FDA and European Medical Agency (EMA) as maintenance treatment for mUC after at least disease stability with platinum-based chemotherapy [7,8].

Our review aims to summarize the evidence of ICIs efficacy in metastatic urothelial BCa, for exploring clinical implications and future development in this field.

## 2. Materials and Methods

We searched PubMed, EMBASE, and Cochrane databases, and abstracts from international conferences (e.g., ASCO, ESMO, ASCO GU). The terms (“metastatic urothelial cancer” OR “advanced urothelial cancer” OR “metastatic urothelial carcinoma” OR “advanced urothelial carcinoma”) AND (“immune checkpoint inhibitor” OR “ICI” OR “avelumab” OR “nivolumab” OR “atezolizumab” OR “pembrolizumab” OR “durvalumab” OR “tremelimumab” OR “ipilimumab” OR “anti PD1” OR “anti PD-L1” OR “anti CTLA4”) were used. Papers published in peer-reviewed journals and conference abstracts in the English language up to September 2021 were selected. We included clinical trials, whereas pre-clinical and animal studies, reviews, letters, and personal opinions were excluded.

A total of 18 studies were included in our review.

## 3. Results

BCa was historically the first malignancy in which immunotherapy showed efficacy: It was 1976 when Bacillus Calmette-Guérin (BCG) showed efficacy in non-muscle-invasive BCa (NMIBC), and even now this treatment is used [21]. In advanced disease, ICIs were initially employed in the platinum-progressing setting, with better response rates and survival results than chemotherapy [22,23,24,25,26,27,28,29,30,31,32]. Subsequently, ICIs were studied in first-line therapy as an alternative to chemotherapy in untreated cisplatinum-ineligible patients or in combination with chemotherapy or antibody-drug conjugates (ADC) [33,34,35,36,37,38,39]. Most recently, ICIs have been used as maintenance in patients who did not progress to platinum-containing chemotherapy [40,41] (Table 1).

### 3.1. Second-Line Treatment

Over the last five years, atezolizumab, pembrolizumab, nivolumab, durvalumab, and avelumab were assessed for safety and efficacy over chemotherapy in BCa patients progressing to platinum-based therapy.

#### 3.1.1. Atezolizumab

Atezolizumab is a humanized IgG1 monoclonal antibody directed against PD-L1. It was the first FDA-approved ICI for advanced BCa in 2016, after publication of the results of the phase II single-arm two-cohort IMvigor210 study. Cohort 2 included patients who progressed to platinum-based chemotherapy. Atezolizumab 1200 mg q3w was administered to 310 patients, reaching an ORR (primary endpoint) of 14.8% (95% confidence interval (CI), 11.1–19.3%; *p* = 0.0058), with 5% of complete responses (CR) after 1 year of follow-up. Patients were also evaluated for PD-L1 expression on immune cells with a cut-off of 5% using a Ventana platform. In patients with PD-L1 ≥5%, ORR was 26%, compared to 9.5% of patients with PD-L1 <5%. After a median follow-up of 11.7 months, 84% of patients were still responding to atezolizumab. Median OS (mOS) was 7.9 mos (95% CI, 6.6–9.3 mos). 1 y-OS was 37% and 2 y-OS was 23%. Median PFS (mPFS) was 2.7 mos (95% CI, 2.1–4.2 mos). ≥G3 adverse events (AEs) occurred in 16% of patients (most commonly diarrhea, fatigue, nausea, and pruritus), and ≥G3 immune-related AEs (irAEs) occurred in 5% (more frequently, pneumonitis, and elevated levels of liver enzymes) [22]. Luminal cluster II of The Cancer Genome Atlas (TCGA) classification demonstrated the highest ORR (34%; *p* = 0.0017). Of note, this subgroup had transcriptional signatures indicating the presence of activated T effector cells [22,42]. In a post-progression analysis including 220 patients that continued atezolizumab beyond the response evaluation criteria in solid tumors (RECIST) progression and until loss of clinical benefit, an mOS of 8.6 months was reported [43].

In the phase III IMvigor211 trial, 931 patients were randomized to atezolizumab vs. chemotherapy. OS was hierarchically tested as the primary endpoint, starting from PD-L1 positive patients. The study did not meet its primary endpoint, as mOS was 11.1 vs. 10.6 mos (HR = 0.87, 95% CI, 0.63–1.21; *p* = 0.41). However, atezolizumab showed a more manageable safety profile, with ≥G3 AEs in 20% of patients (vs. 43% of chemotherapy) [23].

#### 3.1.2. Pembrolizumab

Pembrolizumab is a humanized IgG4 antibody binding PD1. In the KEYNOTE-012 phase Ib study, pembrolizumab was administered at the dosage 10 mg/kg q3w. Patients were included if at least 1% of PD-L1 was detected in tumor cells or stroma: 33 patients were treated in the mUC cohort. ORR was 26% (95% CI, 11–46%), with a good safety profile, as only 15% of patients developed ≥G3 AEs [24].

In the phase III KEYNOTE-045 study, 542 patients who progressed to first-line platinum-based therapy were randomized to pembrolizumab 200 mg q3w versus chemotherapy at investigators’ choice (paclitaxel, docetaxel, vinflunine). OS and PFS—co-primary endpoints—were assessed in the overall population and PD-L1^+^ patients (defined as combined positive score (CPS) ≥10%). Compared to chemotherapy, pembrolizumab reached a longer OS (10.3 mos vs. 7.4 mos; HR = 0.73, 95% CI, 0.59–0.91, *p* = 0.002). The OS improvement was confirmed in the PD-L1^+^ population (8.0 vs. 5.2 mos; HR = 0.57, 95% CI, 0.37–0.88, *p* = 0.005). However, no PFS differences were detected (HR = 0.98, *p* = 0.42 in the overall population; HR = 0.89, *p* = 0.24 in PD-L1^+^ patients). Pembrolizumab was better tolerated than chemotherapy, with AEs occurring in 60.9% vs. 90.2% of patients, and ≥G3 AEs in 15.0% vs. 49.4%. ORR was 21.1% in the pembrolizumab group and 11.4% in the chemotherapy group (*p* = 0.001). mDOR was not reached (NR) with pembrolizumab, whereas it was 14.1 months with chemotherapy; mPFS: 2.2 vs. 3.3 months (*p* = 0.41). Sixty-eight percent of responses to pembrolizumab were ongoing at 12 months [25]. Moreover, after two years of follow-up, 1 y- and 2 y-OS were higher with pembrolizumab (44.2% and 26.9%) than with chemotherapy (29.8% and 14.3%) [44]. Five years of follow-up confirmed the efficacy of pembrolizumab over chemotherapy. Moreover, 32.8% of responses were ongoing [45].

#### 3.1.3. Nivolumab

Nivolumab is an anti-PD1 fully-humanized IgG4 antibody. In the phase II CheckMate 275 study, 265 platinum-progressing mUC patients received nivolumab 3 mg/kg q2w. The primary endpoint was ORR in all treated patients and PD-L1^+^ patients (defined as PD-L1 ≥5%, and after an amendment, ≥1%; PD-L1 expression was assessed on tumor cells through the Dako system); secondary endpoints were PFS and OS. ORR was 19.6% (95% CI, 15.0–24.9%) and did not correlate with PD-L1 expression, as it was 23.8% for PD-L1 ≥1% and 16.1% for PD-L1 <1%. However, a longer OS was reached by PD-L1^+^ patients (11.3 vs. 5.9 mos of PD-L1^−^). Eighteen percent of patients developed ≥G3 AEs, of which diarrhea was the most frequent. Higher interferon (IFN)-gamma signature correlated with nivolumab response (*p* = 0.0003), and it was more expressed in basal 1 subtype of TCGA classification, which also had the highest proportion of responders [26]. After three years of follow-up, mPFS was 1.9 mos, mOS 8.6 mos, higher tumor mutational burden (TMB) was associated with better survival and response rates (*p* < 0.05), having a predictive role for ICIs if combined to PD-L1 [46].

The phase I/II study CheckMate 032 included mUC patients treated with nivolumab 3 mg/kg q2w (NIVO3) or nivolumab 3 mg/kg plus ipilimumab 1 mg/kg q3w for four doses (NIVO3 + IPI1) or nivolumab 1 mg/kg plus ipilimumab 3 mg/kg for 4 doses (NIVO1 + IPI3) followed nivolumab 3 mg/kg q2w. Seventy-eight patients were included in the NIVO3 cohort. ORR—the primary endpoint—was 24.4% (95% CI, 15.3–35.4%). mOS was 9.7 mos (95% CI, 7.3–16.2 mos), mPFS 2.8 mos (95% CI, 1.5–5.9). There was no difference in ORR among PD-L1 positive (24.0%) and negative patients (26.2%). mOS was 16.2 mos in PD-L1^+^ versus 2.8 mos in PD-L1^−^ patients. Ten-point-three percent of patients experienced an ≥G3 AE, leading to treatment discontinuation in 2.6% of cases and two deaths [27]. 1 y- and 2 y- PFS and OS rates were similar between the subgroups, independently from PD-L1 status [47]. In the NIVO3 + IPI1 cohort 104 patients were treated, and there were 92 in the NIVO1 + IPI3 cohort. At the extended follow-up of 3 years, ORR were 25.6%, 26.9%, and 38% in the NIVO3, NIVO3 + IPI1, and NIVO1 + IPI3 cohorts, respectively. In the NIVO1 + IPI3 arm, ORR ranged from 23.8% for PD-L1^−^ to 58.1% for PD-L1^+^ patients. mDOR was similar irrespective of PD-L1 in the three cohorts. mPFS was 2.8, 2.6, and 4.9 mos in the NIVO3, NIVO3 + IPI1, and NIVO1 + IPI3 arms. PD-L1^−^ patients reached mOS of 14.0, 7.4, and 14.9 mos in the NIVO3, NIVO3 + IPI1, and NIVO1 + IPI3 arms; PD-L1^+^ patients had mOS of 12.9, 10.8, and 24.1 months. The three cohorts registered ≥G3 AEs in 26.9%, 30.8%, and 39.1% of patients, respectively [28].

#### 3.1.4. Durvalumab

Durvalumab is a human monoclonal IgG1 antibody directed against PD-L1. In the phase I/II NCT01693562 trial (Study 1108) of durvalumab 10 mg/kg q2w, 61 patients of the urothelial BCa cohort reached an ORR of 31.0%, ranging from 0% in PD-L1 negative (<25% on tumor or immune cells using the Ventana test) to 46.4% in PD-L1 positive (≥25%) patients. The 12 wks disease control rate (DCR)—secondary endpoint—was 57.1% and 28.6% in the PD-L1 positive and PD-L1 negative patients, respectively [29]. A total of 191 patients with mUC progressive to previous chemotherapy then received durvalumab. The ORR was 17.8% (95% CI, 12.7–24.0%) and there were seven CR, regardless of PD-L1 expression. mDOR was NR. Median time to response was 1.41 months, mPFS was 1.5 months, and mOS was 18.2 months. ≥G3 AEs were observed in 6.8% of patients, and ≥G3 irAEs in 2.1% patients, leading to treatment discontinuation in 1.6% and two deaths [30].

#### 3.1.5. Avelumab

Avelumab is an anti-PD-L1 IgG1 antibody. It blocks the interaction between PD1 and PD-L1 but not between PD1 and PD-L2 and induces antibody-dependent cell-mediated cytotoxicity (ADCC) via NK cells [48,49]. In the UC cohort of the phase Ib JAVELIN Solid Tumor, 249 patients received avelumab 10 mg/kg q2w. PD-L1 was evaluated with the Dako system: PD-L1 positive patients had ≥5% expression on tumor cells. Among the platinum-progressing patients, ORR was 17% [31]. Forty-four patients of the expansion cohort were treated with avelumab, reaching an ORR of 18.2% (95% CI, 8.2–32.7%), an mPFS of 11.6 weeks, and an mOS of 13.7 mos (95% CI, 8.5-not estimable (NE)). The 1 y OS-rate was 54.3%. Responses occurred independently from PD-L1 expression, even though a trend towards longer PFS and OS was observed in case of PD-L1 positivity. >G3 AEs occurred in 6.8% of patients: fatigue, infusion-related reaction, and nausea were the most common [32]. After two years of follow-up, mOS was 7 mos (95% C, 5.9–8.5 mos) with a 2 y OS rate of 20.1% (95% CI, 15.2%–25.4%) and an mDOR of 20.5 mos (95% CI, 9.7 mos-NE). A longer OS was achieved in patients responding within the first three months of therapy (NR, 95% CI 18.9 mos-not evaluable (NE), vs. 7.1 mos, 95% CI 5.2–9.0 mos) [50].

### 3.2. First-Line Treatment

Atezolizumab and pembrolizumab received accelerated FDA approval as first-line treatment of cisplatin-ineligible patients in 2017, based on the results of the two single-arm phase II studies, IMvigor210 (cohort 1, atezolizumab) and KEYNOTE-052 (pembrolizumab). Both studies enrolled cisplatin-ineligible mUC patients, regardless of PD-L1 expression [33,35]. However, after the reports of the phase III studies IMvigor130 and KEYNOTE-361, the indications for atezolizumab and pembrolizumab were revised for PD-L1 positive platinum-ineligible patients [20,34,36]. In 2020, the combination of enfortumab vedotin and pembrolizumab received FDA breakthrough designation as first-line therapy for cisplatin-unfit patients with advanced/metastatic UC [37,38].

#### 3.2.1. Atezolizumab

In cohort 1 of the IMvigor 210 study, 119 untreated cisplatin-ineligible mUC patients received atezolizumab 1200 mg q3w. ORR was 23% (95% CI, 16–31%), but in this cohort, the results were independent of PD-L1 status (28% for patients with PD-L1 ≥5% vs. 21% with PD-L1 <1%). mPFS was 2.7 mos (95% CI, 2.1–5.4 mos). mDOR was NR after two years, and mOS was 16.3 mos (95% CI, 10.4–24.5 mos). In this cohort, 8% of patients interrupted the treatment due to AEs, and one treatment-related death was recorded. No differences were observed across TCGA subgroups. Patients with higher TMB had longer OS [33,42].

The subsequent phase III IMvigor130 study included 1312 untreated mUC patients. PD-L1 expression was not an inclusion criterion; however, a stratification by PD-L1 on immune cells was conducted (Ventana SP142 IHC: IC0 = 0%, IC1 = 1–5%, ICI2/3 >5%). Patients were randomized into three groups: group A received atezolizumab plus platinum-based chemotherapy; group B received atezolizumab alone (1200 mg q3w); group C received platinum-based chemotherapy plus placebo (PBO). The co-primary endpoints included PFS and OS group A vs. C, and OS group B vs. C (to be tested in case of A vs. C statistic significance). mPFS was 8.2 mos in group A vs. 6.3 mos in group C (HR = 0.82; 95% CI, 0.70–0.96; *p* = 0.007). mOS was 16.0 months in group A vs. 13.4 months in group C (HR = 1.02; 95% CI, 0.83–1.24). Therefore, the trial failed to meet the OS endpoint. However, a more robust OS benefit was achieved in the subgroup of patients that received cisplatin than carboplatin. Given the hierarchical testing study procedure, the comparison of atezolizumab alone versus chemotherapy was never made. Among patients with PD-L1 ≥5%, mOS was longer with atezolizumab (NE) than with chemotherapy alone (17.8 mos; HR = 0.68, 95% CI, 0.43–1.08). However, PD-L1 negative patients did not benefit from atezolizumab compared to chemotherapy (mOS 13.5 vs. 12.9 mos; HR = 1.07; 95% CI, 0.86–1.33) [34]. High PD-L1 and TMB were associated with favorable OS with atezolizumab monotherapy [51]. The combination of atezolizumab and chemotherapy determined the worst toxicity, as 96% of patients developed AEs, of which 81% were G3–G4. Atezolizumab alone determined 60% of AEs, of which 15% were severe; chemotherapy led to 81% of AEs (35% severe AEs). Treatment withdrawal occurred in 34% of patients treated with chemotherapy plus atezolizumab, 6% of patients receiving atezolizumab, and 34% of chemotherapy patients. Treatment-related deaths were double in group A compared to groups B and C (2% vs. 1% vs. 1%) [34].

#### 3.2.2. Pembrolizumab

In the phase II single-arm KEYNOTE-052 study, pembrolizumab 200 mg q3w was administered to 317 untreated cisplatin-unfit mUC patients. ORR—the primary endpoint—was 28.6% (95% CI, 24.1–33.5%). Nine percent of CR and 20% PR were reported after two years of follow-up. mDOR was 30.1 months, mOS was 11.3 mos, and over 50% of patients had durable responses ≥24 months. PD-L1 was assessed as CPS: among patients with CPS ≥10, ORR was 47.3% and mOS 18.5 months. ≥G3 AEs were reported in 16% of patients [35,52].

In the phase III KEYNOTE-361 study, 1010 patients were randomized 1:1:1 to pembrolizumab q3w versus pembrolizumab plus chemotherapy (cis/carboplatin + gemcitabine) versus chemotherapy alone. Patients were enrolled regardless of PD-L1 staining, but CPS score was evaluated (cut-off: 10%). The co-primary endpoints OS and PFS were not met. In the three groups, mPFS was 8.3, 3.9, and 7.1 months (HR for pembrolizumab plus chemotherapy vs. chemotherapy 0.78, 95% CI, 0.65–0.93; *p* = 0.0033), mOS were 17.0, 15.6, and 14.3 months (HR pembrolizumab plus chemotherapy versus chemotherapy 0.86, 95% CI, 0.71–1.02; *p* = 0.0407); ORR were 54.7%, 30.3%, and 44.9%, respectively. After these findings, the indications for pembrolizumab as first-line treatment of BCa were revised. Seventy-five percent of patients treated with pembrolizumab plus chemotherapy developed ≥G3 AEs, versus 17% of patients treated with pembrolizumab alone and 33% of patients receiving chemotherapy. Discontinuation occurred in 31% of patients after pembrolizumab plus chemotherapy, as opposed to 16% of patients treated with pembrolizumab and 18% of patients receiving chemotherapy. Anemia was the most common AE after chemotherapy; pruritus, rash, and fatigue were typical of pembrolizumab [36].

ADCs consist of monoclonal antibodies conjugated with cytotoxins: after ADC is recognized on the cell surface, the complex is internalized and the specific cytotoxin released. Enfortumab vedotin is an ADC targeting nectin-4, a surface molecule involved in cellular adhesion, expressed by tumor cells, mUC included. The combination of enfortumab vedotin and pembrolizumab in the first-line setting of mUC was investigated in the phase Ib/II trial EV-103: among 45 cisplatinum-ineligible patients, an ORR of 73.3% and an mPFS of 12.3 months were recorded [37]. After two years of follow-up, mDOR was 25.6 months, OS-rate was 56.3%, mPFS was 12.3 months, and mOS was NR. Fatigue, peripheral sensory neuropathy, and alopecia presented in around half of the patients [38]. Based on these results, the FDA granted breakthrough therapy approval for the combination of enfortumab vedotin and pembrolizumab as first-line therapy of cisplatin-unfit patients with advanced/metastatic UC.

#### 3.2.3. Durvalumab

The phase III DANUBE trial assessed the combination of durvalumab with the CTLA4 inhibitor tremelimumab, compared to chemotherapy, in the first-line setting of mUC. One-thousand and thirty-two patients were randomized 1:1:1 to receive durvalumab (1500 mg q4w), or durvalumab plus tremelimumab (75 mg q4w up to 4 doses, followed by durvalumab alone), or chemotherapy. The primary endpoint OS was tested in the PD-L1 positive patients comparing durvalumab to chemotherapy, and in the overall population comparing durvalumab plus tremelimumab to chemotherapy. PD-L1^+^ patients reached an mOS of 14.4 months with durvalumab, vs. 12.1 months with chemotherapy (HR = 0.89; 95% CI, 0.71–1.11; *p* = 0.30). The other co-primary endpoint in the overall population was not met either: mOS was 15.1 mos with durvalumab plus tremelimumab, vs. 12.1 months with chemotherapy (HR = 0.85; 95% CI, 0.72–1.02; *p* = 0.075). Chemotherapy was far less tolerated than immunotherapy; in fact, AEs occurred in 60% of patients. However, the addition of tremelimumab to durvalumab almost doubled the AEs rate (27% vs. 14%). Each of the three groups registered a treatment-related death [39].

### 3.3. Switch Maintenance Immunotherapy after First-Line Chemotherapy

Switch-maintenance therapy aims to prolong the strength and endurance of chemotherapy-induced responses through ICIs [53]. Results of avelumab and pembrolizumab trials are available.

#### 3.3.1. Avelumab

In the phase III trial JAVELIN Bladder 100, patients who had achieved a partial/complete response or stable disease after first-line chemotherapy with cis- or carboplatin plus gemcitabine (at least four cycles) were randomized 1:1 to receive avelumab maintenance plus best-supportive care (BSC) vs. BSC alone, after a treatment-free interval (TFI) of 4–10 weeks. Seven hundred patients were included. The primary endpoint—OS—was met in the overall population and among PD-L1^+^ patients: in the overall population, avelumab prolonged mOS to 21.4 vs. 14.3 months of chemotherapy (HR = 0.69; 95% CI, 0.56–0.86; *p* = 0.001); in PD-L1^+^ patients, mOS was NR in the avelumab group, vs. 17.1 in the BSC group (HR = 0.56; 95% CI, 0.40–0.79; *p* < 0.001). Results were independent of platinum compounds, response magnitude to previous chemotherapy, age, sites of metastases, renal function [40]. Of note, a higher proportion of patients in the avelumab group received subsequent therapies than the BSC group (53% vs. 9%). AEs were recorded in 17% of patients in the avelumab group (vs. 0% of BSC), with increased pancreatic enzymes being the most common. Avelumab induced irAEs in 29% of patients, including 7% ≥G3 AEs [40,54].

#### 3.3.2. Pembrolizumab

In the randomized phase II GU14-182 trial, 107 UC patients that did not progress to first-line platinum-based chemotherapy were randomized to maintenance with pembrolizumab vs. PBO. The primary endpoint—PFS—was prolonged with pembrolizumab (5.4 vs. 3.0 mos, HR = 0.65; *p* = 0.04) but no OS advantage (secondary endpoint) was achieved (22.0 vs. 18.7 mos; HR = 0.91, 95% CI, 0.52–1.59; *p* = 0.75). PD-L1 (defined as CPS ≥10, Dako platform) did not impact survival. As permitted by the trial, 52% of patients crossed to pembrolizumab. This may have altered the significance of OS results. ≥G3 AEs occurred in 59% of patients treated with pembrolizumab vs. 39% in the PBO arm [41].

## 4. Discussion

BCa is a common tumor of the genito-urinary tract. Fortunately, over half of the cases are diagnosed as NMIBC, which has a good survival, whereas in the metastatic setting, life expectancy is short [1,2,3]. Platinum-based regimens remain a standard approach for first-line therapy [9,10,11,12]. ICIs approval represented a turning point in terms of response and survival rates. Indeed, after the enthusiastic result of second-line ICIs, immunotherapy suffered a setback in the first line, with dismal responses and survival rates leading to approval revisions by the regulatory agencies [20,22,23,24,25,26,27,28,29,30,31,32,33,34,35,36,37,38,39]. The evidence of immunological effects of many chemotherapeutic agents gave the rationale for chemo-immunotherapy combinations and ICIs maintenance after chemotherapy. Indeed, chemotherapy induces the release of neo-antigens via cell apoptosis, favoring immune response [55]. Cisplatin exerts an additional series of immunomodulant effects: upregulation of major histocompatibility complex (MHC) class I on tumor and antigen-presenting cells (APCs), recruitment and proliferation of effector T-cells and APCs, PD-L1 upregulation, IFN production, and downregulation of immunosuppressive myeloid-derived suppressor cells (MDSC) and Tregs [56,57,58,59]. Gemcitabine also has immunomodulant properties: MDSC and Treg depletion, accumulation of tumor-associated macrophages (TAMs) and shifting them towards immunostimulant activity, MHC class I upregulation, APC induction, and NK cells activation [60,61].

Nowadays, choosing the right timing for starting ICIs can profoundly change patients’ survival. As ICIs monotherapies do not represent an advantage, having reached a maximum OS of 12 months, they might be better used for PD-L1^+^ platinum-ineligible patients. The combination of anti-PD1/PD-L1 and chemotherapy or anti-CTLA-4 agents has achieved longer OS (15–17 months), but without statistical significance. Promising results could come from the combination of pembrolizumab and enfortumab vedotin in cis-unfit patients, as mPFS is 12 months, but mOS has not yet been reached. Indeed, as avelumab maintenance adds 21.4 months of mOS to the 4–6 months of first-line platinum-based chemotherapy—in stable/responder patients—this represents the most beneficial combination for mUC patients. Administered after progression to first-line chemotherapy, ICIs add 8–15 months, for a total survival of 16–21 months. (Figure 1).

Moreover, with ICIs maintenance, the number of BCa patients receiving immunotherapy during their treatment history has increased, including the majority of those interdicted from second-line therapies due to worsening clinical conditions after progression to first-line therapy. Nevertheless, at least two questions remain. First of all, what the proper timing of maintenance starting and the optimal number of cycles of chemotherapy could be: a recent subgroup analysis of JAVELIN Bladder 100 showed that the survival benefit was independent of TFI and time from chemotherapy end to avelumab start [62]. Secondly, the sequencing strategy after progression to ICIs maintenance should be investigated: currently, possible options are represented by erdafitinib for patients with fibroblast-growth factor receptor (FGFR) 2/3 alterations, enfortumab vedotin, chemotherapy with platinum rechallenge (in patients with long TFI), and other agents (docetaxel, or vinflunine in Europe) [7,8].

Combinations with other agents are under evaluation, such as the dual anti-PD1/anti-CTLA4 inhibitor, and the triplet pembrolizumab + enfortumab vedotin with or without chemotherapy. Another potentially effective ICIs combination is with anti-angiogenic and tyrosine-kinase inhibitors (TKI), which modulate the tumor microenvironment (TME) [63]. The dose-expansion cohort of the NCT02496208 phase I study evaluated the combination of Cabozantinib and Nivolumab alone or plus Ipilimumab in many metastatic GU tumors: 15 patients of mUC cohort reached ORR 38.5%, mOS was 25.4 months, and mPFS was 5.1 months. However, ≥G3 AEs occurred in 75% and 87% of patients treated with dual and triple therapy, respectively [64,65]. In the second line, the combination of pembrolizumab with ramucirumab achieved an ORR of 13% [66]. Other agents, such as vaccines, epigenetic modulators (e.g., Enhancer of zeste homolog 2 (EZH2) inhibitors), ADC, and poly-ADP ribose polymerase (PARP) inhibitors, represent potentially effective ways to improve ICIs efficacy [67,68]. Preliminary results of phase I/II trials of ICIs combined with anti-FGFR agents show an ORR of 11–30% in pre-treated patients, reaching 54% in the first-line case [69,70,71,72] (Table 2).

Biomarkers with a predictive role for ICIs in BCa are warranted. Historically, PD-L1 is the first biomarker to which such a role was attributed. Effectively, PD-L1 expression has been found in around 20–30% of BCa, often associated with increased pathologic stage and worse survival [73,74]. However, the predictivity of PD-L1 expression has been inconsistent across the trials conducted in BCa. This is partially due to the lack of standardized methods for defining PD-L1 positivity among the different studies, PD-L1 detection methods, the cut-off for positivity, and choice of tumor cells, immune cells, or both, for detection (Table 1). Effectively, in Study 1108, when PD-L1 positive tumor cells were considered (Ventana SP263 system), an ORR of 46.7% was reached with durvalumab, versus 22.2% in case of PD-L1 negativity. By PD-L1 expression on immune cells, ORR was 55.6% for PD-L1 positive versus 12.5% for PD-L1 negative subjects. When both the tumor and immune cells expression were considered, the ORR was 0 in patients without PD-L1 expression and rose to 46.4% when, alternatively, the tumor or the immune cells were PD-L1 positive [29]. In the IMvigor210 trial, PD-L1 positivity correlated with ORR to atezolizumab in second but not in first line, and in IMvigor310 atezolizumab determined a higher mOS among PD-L1 positive than negative patients (both studies employed Ventana SP142 platform) [22,33]. On the contrary, nivolumab did not improve ORR in PD-L1 positive compared to negative patients in CheckMate-275 and -032 studies (assessing PD-L1 through Dako 28.8 system), but in CheckMate-032 an OS prolongation was achieved in case of PD-L1 expression [26,27,28]. Responses to avelumab and pembrolizumab were independent from PD-L1 status (Dako 73–10 and 22C3, respectively) [24,25,31,32,35,36]. Albeit a greater effect in PD-L1 positive patients (Ventana SP263), avelumab maintenance improved survival also in PD-L1 negative patients [40,75]. Baseline tumor dimensions and high PD-L1 were positively associated with durvalumab response and OS in Study 1108, as if smaller tumor size permitted higher penetration of immune cells [76]. Besides surface expression, post-translation modifications of PD-L1 (such as N-glycosylation) should be better enquired, as they are reported to influence PD1 binding, receptor stability, and detectability [77].

Together with PD-L1 expression, novel biomarkers should be included in a multi-marker classification, as it seems unlikely that a single biomarker would effectively guide all treatment decisions [75,78]. V-domain IG suppressor of T-cell activation (VISTA) is a ligand for APCs of B7 family, expressed in myeloid cells, granulocyte, and T-cells, acting as a negative regulator of T-cells activation, proliferation, and cytokine production. In primary BCa, VISTA expression was significantly associated with PD-L1 expression. Moreover, the VISTA expression on TILs correlated with a shorter disease recurrence. Therefore, VISTA has been purposed as an immunological biomarker for BCa [79].

As in other cancer subtypes, TMB is a reliable biomarker for ICI response in BCa, due to a higher neo-antigens production [22,80]. It became apparent in the IMvigor210 trial, in which responders had a higher TMB than non-responders (12.4 vs. 6.4 per megabase; *p* < 0.0001), that positively correlated with OS [22,33]. Similarly, a high TMB was associated with higher ORR (*p* = 0.002) and longer PFS (3.0 vs. 1.9 mos) in patients treated with nivolumab in the CheckMate 275 study [26]. In JAVELIN Bladder 100, the HR with avelumab maintenance was lower in patients with higher TMB (*p* = 0.26). A possible contribution to increased TMB derives from specific mutations linked to the APOBEC signature, associated with improved OS to avelumab maintenance (*p* = 0.02) [75]. Additionally, for TMB, there is a need for unifying criteria for defining high and low levels and their predictive value.

The last TCGA classification (2017) defined five different subtypes of MIBC, having different prognoses and propensities to respond to chemo/immunotherapy: luminal has the best prognosis, and potential benefits from anti-FGFR therapies; luminal-infiltrated, expressing luminal genes, PD-L1 and CTLA-4, and it has a good response to ICIs; luminal-papillary has both luminal genes and extracellular matrix elements; basal-squamous shows the worst prognosis, basal and immune genes, a good response to chemotherapy, and ICIs; neuronal is characterized by neural differentiation [78]. In the IMvigor210 and CheckMate 275 studies, TCGA classification was investigated as a possible predictive biomarker for ICIs: in the former, a higher response rate was observed among patients with the luminal-infiltrated subtype (34%, *p* = 0.0017) treated with atezolizumab, even if PD-L1 positive cells were most frequently found in the basal subtype (60%); in the latter, the highest response rate to nivolumab was obtained in the basal subtype (30%), followed by the luminal-infiltrated (25%) [22,26]. Indeed, BCa is heterogeneous, and different molecular subtypes could coexist in the same tumor, limiting the precision of molecular classification and the predictivity for ICIs response [81]. More recently, molecular classifications have also focused on TME, and gene-expression profiling (GEP) analysis distinguished four immune subtypes of BCa (C1–C4), which differ for behavior and therapy sensitivity, ranging from C2 subtype—immune-infiltrated, associated with longer OS and PFS, and sensitive to ICIs and chemotherapy, to C4 subtype—deprived of CD8^+^ T-cells, with the worst prognosis, inadequate response to ICIs, sensitive to chemotherapy [82,83]. In CheckMate 275, KEYNOTE-052, Study 1108, and JAVELIN Bladder 100, gene-expression profiling (GEP) analysis showed that expression of immune genes related to γ-IFN correlated with nivolumab, pembrolizumab, durvalumab, and avelumab responses, confirming this group of genes as a critical pathway for innate and adaptive immune responses [26,30,35,75]. IFN-mediated responses are also induced after DNA-damage response (DDR) gene mutations, which therefore could predict ICIs response [75,84,85]. Effectively, BCa—mainly basal squamous types—displays a genetic complexity associated with DDR genes, increased TILs, and enhanced platinum and immune responsiveness [85,86]. As proposed in JAVELIN Bladder 100, a contribution of both innate and adaptive immunity (mainly represented by NK cells and CD^+^ cells) could predict avelumab survival [75].

Adverse events represent another matter of interest. ICIs have shown manageable safety profiles as single agents, in line with anti-PD1/PD-L1 treatments in other tumor subtypes [87]. Typical immune-related AEs reported in the studies were pneumonitis, rash, and elevated levels of liver enzymes, requiring specific management by the clinicians [87,88]. However, ICIs were far less tolerable in combination with chemotherapy. In fact, as reported in the studies, the rates of all-grades of AE were much higher than ICIs alone (up to 96% for AEs, and 81% for severe AEs), leading to treatment withdrawal in up to one out of three patients. Moreover, in addition to the immune-related AEs, patients also developed typical chemo-related AEs, such as neutropenia, anemia, and thrombocytopenia, resulting in more difficult monitoring and management [88]. In this regard, further information will be provided from real-world data.

## 5. Conclusions

ICIs have significantly reshaped the BCa treatment paradigm, improving life expectancy for metastatic patients. Compared to chemotherapy, the benefit in the second-line setting is unquestionable. Differently, upfront immunotherapy cannot handle cisplatin results, and safety is worse for immunotherapy and chemotherapy combinations. Early ICIs start as maintenance therapy after first-line platinum-based chemotherapy ensures the highest survival rates and allows the highest numbers of subjects to receive ICIs during their clinical history, representing the best choice for improving patients’ survival. The combination of ICIs and ADCs seems another very effective strategy, with ongoing first-line OS investigation. Sequencing strategies are warranted to maintain the highest survival rates. So far, single biomarkers fall short in helping to select patients for ICIs, whereas multimarker classifications could allow the design of tailored trials.

## Figures and Tables

**Figure 1 biomedicines-10-00411-f001:**
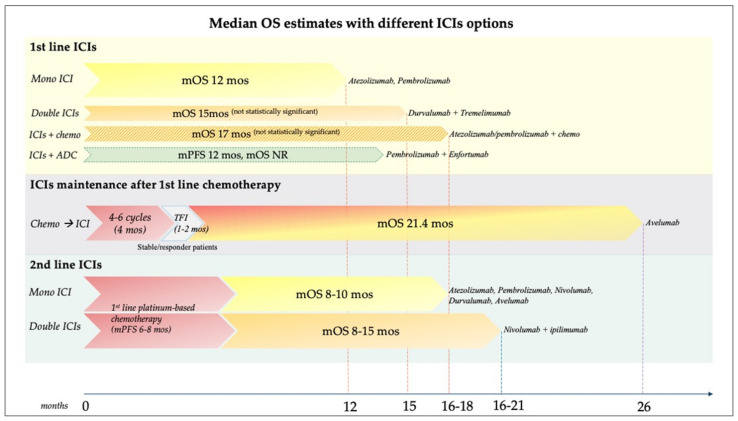
Median overall survival (OS) estimates with different ICI options. In first-line: OS around 12 months with ICI monotherapies; OS 15–17 months with anti-PD1/PD-L1 plus chemotherapy or anti-CTLA-4 (not statistically significant in the studies); pembrolizumab plus Enfortumab vedotin has not reached mOS, but seems promising as mPFS is 12 months. Avelumab maintenance adds 21.4 months of mOS to the 4–6 months of first-line chemotherapy in stable/responder patients, for a total of 26 months, which so far represents the longest survival for metastatic BCa patients. In second-line: ICIs add additional 8–15 months to chemotherapy, for a total OS of 16–21 months. ADC: antibody-drug conjugate; BCa: bladder cancer; CTLA4: cytotoxic T-lymphocyte-associated protein 4; ICI: immune checkpoint inhibitor; mOS: median overall survival; mPFS: median progression-free survival; NR: not reached; PD1: programmed death 1; PD-L1: programmed death-ligand 1; TFI: treatment-free interval.

**Table 1 biomedicines-10-00411-t001:** Clinical trials of ICIs in metastatic BCa.

Line	Trial	Phase	Nr. Patients	ICI Therapy	Control Group	Primary Endpoint	PD-L1 Cut-Off, Cell Types (Detection Platform)	Efficacy Outcomes
2	IMvigor 210 (Cohort 2) [22]	II	310	Atezolizumab 1200 mg q3w	-	ORR	5%, IC (Ventana SP142)	ORR 14.8% mOS 7.9 mos mPFS 2.1 mos
	IMvigor 211 [23]	III	931	Atezolizumab 1200 mg q3w	CT	OS	5%, IC (Ventana SP142)	mOS 11.1 vs. 10.6 mos (PD-L1^+^ patients)
	KEYNOTE-012 (mUC cohort) [24]	Ib	33	Pembrolizumab 10 mg/kg q2w	-	ORR, safety	1%, TC (Dako 22C3) *	ORR 26%
	KEYNOTE-045 [25]	III	542	Pembrolizumab 200 mg q3w	CT	OS, PFS in overall population and PD-L1^+^	CPS ≥ 10, TC and IC (Dako 22C3)	mOS 10.3 vs. 7.4 mos ORR 21.1% vs. 11.4% mDOR NR vs. 14.1 mos mPFS 2.1 vs. 3.3 mos
	CheckMate 275 [26]	II	265	Nivolumab 3 mg/kg q2w	-	ORR (overall, PD-L1^+^)	5%, amended to 1%, TC (Dako 28.8)	ORR 19.6% mPFS 1.9 mos mOS 8.6 mos
	CheckMate 032 [27,28]	I/II	78	Nivolumab 3 mg/kg q2w (NIVO3)	-	ORR	1%, TC (Dako 28.8)	ORR 25.6% mOS: 9.7 mos mPFS 2.8 mos
			104	Nivolumab 3 mg/kg + Ipilimumab 1 mg/kg (NIVO3+IPI1)				ORR 26.9% mPFS 2.6 mos mOS: 7.4 mos (PD-L1^−^), 10.8 mos (PD-L1^+^)
			92	Nivolumab 1 mg/kg + Ipilimumab 3 mg/kg (NIVO1 + IPI3)				ORR 38% mPFS 4.9 mos mOS: 14.9 mos (PD-L1^−^), 24.1 mos (PD-L1^+^)
	NCT01693562 (UBC cohort) [29]	I/II	61	Durvalumab 10 mg/kg q2w	-	Safety	25%, TC/IC (Ventana SP263)	ORR 31.0%
	STUDY 1108 [30]	I/II	191	Durvalumab 10 mg/kg q2w	-	Safety, ORR	25%, TC/IC (Ventana SP263)	ORR 17.8% mPFS 1.5 mos mOS 18.2 mos
	JAVELIN Solid Tumor [31]	Ib	249	Avelumab 10 mg/kg q2w	-	Safety	5%, TC (Dako 73–10)	ORR 17%
	JAVELIN (mUC expansion cohort) [32]	Ib	44	Avelumab 10 mg/kg q2w	-	Safety	5%, TC (Dako 73–10)	ORR 18.2% mPFS 11.6 wks mOS 13.7 mos
1	IMvigor 210 (cohort 1) [33]	II	119 (cis-unfit)	Atezolizumab 1200 mg q3w	-	ORR	5%, IC (Ventana SP142)	ORR 23% mPFS 2.7 mos mOS 16.3 mos
	IMvigor 130 [34]	III	1312	Group A: Atezolizumab + platinum-based CTGroup B: Atezolizumab (1200 mg q3w)	Group C: CT + PBO	PFS, OS (A vs. C), OS (B vs. C if A vs. C was positive)	1% (IC1), 5% (IC2/3) (Ventana SP142)	mPFS 8.2 (A) vs. 6.3 (C) mos mOS 16.0 (A) vs. 13.4 (C) mos
	KEYNOTE-052 [35]	II	317 (cis-unfit)	Pembrolizumab 200 mg q3w	-	ORR	CPS ≥10%, TC and IC (Dako 22C3)	ORR 28.6% mOS 11.3 mos
	KEYNOTE-361 [36]	III	1010	Pembrolizumab (P), Pembrolizumab + CT (P + C)	CT (C)	OS, PFS (starting from P + C vs. C)	CPS ≥10%, TC and IC (Dako 22C3)	mPFS 8.3 (P + C), 3.9 (P), 7.1 (C) mos mOS 17 (P + C), 15.6 (P), 14.3 (C) mos ORR 54.7% (P + C), 30.3% (P), 44.9% (C)
	EV-103 [37,38]	Ib/II	45 (cis-unfit)	Pembrolizumab + Enfortumab vedotin 1.25 mg/kq d1,8 q3w	-	Safety	NA	ORR 73.3% mPFS 12.3 mos mDOR 25.6 mos mOS NR after 2 y
	DANUBE [39]	III	1032	Durvalumab 1500 mg q4w (D), or durvalumab + tremelimumab (75 mg q4w) (D + T)	CT	OS PD-L1^+^ (D vs. CT), OS overall (D + T vs. CT)	25% TC or 25% IC + 1% TC (Ventana SP263)	mOS 14.4 (D) vs. 12.1 mos (CT) (PD-L1^+^ patients) mOS 15.1 (D + T) vs. 12.1 (CT) mos (overall population)
1M	JAVELIN Bladder 100 [40]	III	700	Avelumab 10 mg/kg q2w	BSC	OS (overall, PD-L1^+^)	25%, TC/IC (Ventana SP263)	mOS 21.4 vs. 14.3 mos
	GU14-182 [41]	II	107	Pembrolizumab 200 mg q3w	PBO	PFS	CPS ≥ 10%, TC and IC (Dako 22C3)	mPFS 5.4 vs. 3.0 mos mOS 22.0 vs. 18.7 mos

* Inclusion criterion. 1M: maintenance after first line; BSC: best supportive care; CPS: combined positive score; CT: chemotherapy; IC: immune cells; mDOR: median duration of response; mOS: median overall survival; mPFS: median progression-free survival; NA: not available; NR: not reached; ORR: overall response rate; PBO: placebo; PD-L1: programmed death-ligand 1; TC: tumor cells; UBC: urothelial bladder cancer.

**Table 2 biomedicines-10-00411-t002:** Ongoing trials of ICIs in mUC.

Trial	Phase	Line of Therapy	ICIs and Combinations (Mechanisms of Action)	Primary Endpoints
NCT03036098 (CheckMate 901)	III	1	Nivolumab + Ipilimumab vs. Nivolumab + SOC (CT) vs. SOC	PFS, OS
NCT03682068 (NILE)	III	1	Durvalumab + CT vs. Durvalumab + Tremelimumab + CT vs. CT	OS (PD-L1 >25%)
NCT04223856 (EV-302)	III	1	Pembrolizumab + Enfortumab vedotin (anti Nectin-4) vs. CT vs. Pembrolizumab + Enfortumab vedotin + CT	PFS, OS
NCT04863885	I/II	1 (cis-unfit)	Nivolumab + Ipilimumab + Sacituzumab govitecan (anti Trop-2)	MTD, ORR
NCT03898180	III	1	Pembrolizumab + Lenvatinib (TKI)	PFS, OS
NCT03534804	II	1 (cis-unfit)	Pembrolizumab + Cabozantinib (TKI)	ORR
NCT03601455	II	1	Durvalumab + Tremelimumab + RT	PFS, safety
NCT03513952	II	1	Atezolizumab + CYT107 (glycosylated recombinant human IL-7)	ORR
NCT03459846 (BAYOU)	II	1	Durvalumab + Olaparib (PARP-inhibitor)/PBO	PFS
NCT03854474	I/II	1	Pembrolizumab + Tazemetostat (EZH2 inhibitor)	ORR
NCT02500121	II	1M	Pembrolizumab (vs PBO)	6 mos PFS rate
NCT04678362	II	1M	Avelumab + Talazoparib (PARP-inhibitor)	PFS
NCT03473756 (FORT-2)	I/II	1 (FGFR-mutant)	Atezolizumab + rogaratinib (anti-FGFR)/PBO	Safety, PFS
NCT03473743 (NORSE)	I/II	1 (FGFR-mtant)	Cetrelimab (anti-PD1) + erdafitinib (anti-FGFR) vs. erdafitinib	Safety, ORR
NCT04045613	I/II	1 (FGFR-mtant)	Atezolizumab + derazantinib (anti-FGFR)	Safety, ORR
NCT04601857	II	1 (FGFR-mutant)	Pembrolizumab + Futibatinib (anti-FGFR)	ORR
NCT03715985	I/II	any	Avelumab OR atezolizumab OR durvalumab OR nivolumab OR pembrolizumab + Personalized neoantigen vaccine	Safety
NCT02643303	I/II	any	Durvalumab + polyICLC (TLR3 agonist) + in-situ vaccination with tremelimumab	24 wks PFS
NCT02897765	I	2	Nivolumab + NEO-PV-01 (vaccine) + polyICLC	Safety
NCT03915405	I	2	Avelumab + KHK2455 (anti-IDO)	Safety
NCT03606174	II	2 *	Nivolumab or pembrolizumab/enfortumab vedotin + sitravanib (TKI)	ORR
NCT04902040	Ib/II	2 *	Avelumab OR atezolizumab OR durvalumab OR nivolumab OR pembrolizumab + plinabulin (anti-angiogenic) and RT	ORR, safety

* After progression to ICI. CT: chemotherapy; EZH2: Enhancer of zeste homolog 2; FGFR fibroblast growth factor receptor-2; IDO: Indoleamine-pyrrole 2,3-dioxygenase; IL: interleukin; M: maintenance; MTD: maximum tolerated dose; ORR: overall response rate; PARP: poly-ADP ribose polymerase; PBO: placebo; PD1: programmed death 1; PD-L1: programmed death-ligand 1; PFS: progression-free survival; RT: radiotherapy; SOC: standard of care; TKI: tyrosine-kinase inhibitor; TLR3: toll-like receptor 3; Trop-2: Tumor-associated calcium signal transducer 2.

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
