# Peer review of "Immune-Checkpoint Inhibitors in Advanced Bladder Cancer: Seize the Day"

_biomedicines, 2022, doi:10.3390/biomedicines10020411_

Round 1

Reviewer 1 Report

The authors present the review on ICIs in advanced BCa. Based on the literature search but not in the systematic setting, they summarized 18 studies focusing on the drug use in monotherapy or combination. Definitely, there is growing hope in ICI due to the suboptimal management of BCa patients, which is represented by poor general survival. Are ICIs truly the game changers? The authors included both second- and first-line settings so as to widen horizon of the options. In the figure 1 they managed to display the real value of the treatment in prolonging patients survival. Second-line treatment has already established position, while initial immunotherapy proved to be inferior when confronted with chemotherapy.

Worth mentioning in the discussion is the AEs management as it can be truly challenging, especially in combination setting.

All the sections (biomarkers, including PDL-1 expression; subclassification importance) were included. As for the biomarkers, it is worth discussing about other possible directions e.g. V-domain immunoglobulin suppressor of T cell activation (VISTA) is an immune checkpoint molecule expressed in hematopoietic cells, granulocytes, macrophages, and monocytes. Some authors (10.1007/s00262-021-02906-7) suggested VISTA may become a potential immunotherapeutic target and immunologic biomarker in bladder cancer.

Author Response

We are truly grateful to the reviewer for their valuable comments.

We updated the safety profiles of single molecules when previously absent. We added a paragraph discussing the adverse events and immune-related safety profiles of single agents and combination with chemotherapy.

We added a paragraph mentioning VISTA among biomarkers discussion.

Reviewer 2 Report

Thank you for sharing this useful review. This review showed the comprehensive knowledge about the current treatment in advanced bladder cancer. There are some minor concerns. Please check the following points.

  1. Figure 1 is useful for treatment strategy. To understand more easily, please add the name of clinical trial and drugs. Please add the Abbreviation definition.
  2. Table 1 is so informative. But, it may be a little difficult to read. Please revise it, especially efficacy outcomes.

Author Response

We are truly grateful to the reviewer for their valuable comments.

We added the names of the drugs in Figure 1, and also the 'Abbreviations definition'.

We revised and simplified Table 1.

Reviewer 3 Report

At the beginning, I would like to congratulate the authors on a very good and complete development of the topic

General comments

The article is well organized and clear. The subject is of interest to the field.

The methods are correctly described and applied. 

The results are clearly presented, and tables and figures are appropriate.

The discussion is balanced and significant.

The conclusions are consistent with the results and discussion.

References are adequate in number and range.

One small remark - with regard to adverse events taking into account the AE of special interest, ie immunotherapy-specific, the paragraph summarizing the adverse effects is missing. When describing individual molecules, toxicities are mentioned, but not in all parts of the manuscript. Systematizing this issue would make the article complete.

Author Response

We are truly grateful to the reviewer for their valuable comments.

We updated the safety profiles of single molecules when previously absent. We added a paragraph discussing the adverse events and immune-related safety profiles of ICIs as single agents and in combination with chemotherapy.